# DJ-1 controls bone homeostasis through the regulation of osteoclast differentiation

Hyuk Soon Kim[1], Seung Taek Nam[1], Se Hwan Mun[1,2], Sun-Kyeong Lee[2], Hyun Woo Kim[1], Young Hwan Park[1], Bokyung Kim[1], Kyung-Jong Won[1], Hae-Rim Kim[3], Yeong-Min Park[1], Hyung Sik Kim[4], Michael A. Beaven[5], Young Mi Kim[6] & Wahn Soo Choi[1]

Receptor activator of NF-kB ligand (RANKL) generates intracellular reactive oxygen species (ROS), which increase RANKL-mediated signaling in osteoclast (OC) precursor bone marrow macrophages (BMMs). Here we show that a ROS scavenging protein DJ-1 negatively regulates RANKL-driven OC differentiation, also called osteoclastogenesis. DJ-1 ablation in mice leads to a decreased bone volume and an increase in OC numbers. In vitro, the activation of RANK-dependent signals is enhanced in DJ-1-deficient BMMs as compared to wild-type BMMs. DJ-1 suppresses the activation of both RANK-TRAF6 and RANK-FcRγ/Syk signaling pathways because of activation of Src homology region 2 domain-containing phosphatase-1, which is inhibited by ROS. Ablation of DJ-1 in mouse models of arthritis and RANKL-induced bone disease leads to an increase in the number of OCs, and exacerbation of bone damage. Overall, our results suggest that DJ-1 plays a role in bone homeostasis in normal physiology and in bone-associated pathology by negatively regulating osteoclastogenesis.

[1] Department of Immunology and Physiology, School of Medicine, Konkuk University, Chungju 380-701, Republic of Korea. [2] Department of Medicine, University of Connecticut Health Center, 263 Farmington Ave, Farmington, CT 06030, USA. [3] Department of Rheumatology, School of Medicine, Konkuk University, Chungju 380-701, Republic of Korea. [4] Department of Toxicology, School of Pharmacy, Sungkyunkwan University, Suwon 440-746, Republic of Korea. [5] Laboratory of Molecular Immunology, National Heart, Lung and Blood Institute, National Institutes of Health, Bethesda, MD 20892, USA. [6] Department of Preventive Pharmacy, College of Pharmacy, Duksung Women's University, Seoul 132-714, Republic of Korea. Correspondence and requests for materials should be addressed to W.S.C. (email: wahnchoi@kku.ac.kr)

The maintenance of bone homeostasis is dependent on the balance of activity of bone-resorbing osteoclasts (OCs) and bone-forming osteoblasts (OBs)[1, 2]. Abnormal bone resorption by OCs results in bone destruction and is characteristic of bone-related diseases such as osteoporosis, Paget's disease of bone and rheumatoid arthritis (RA)[3]. OCs are derived from monocyte/macrophage lineage cells. Formation of functional OCs is dependent on macrophage colony-stimulating factor (M-CSF) and receptor activator of nuclear factor κB (NF-κB) ligand (RANKL)[4, 5]. M-CSF ensures proliferation and survival of the OC precursor cells by acting through its receptor c-Fms to activate mainly Akt and ERK1/2[6]. RANKL promotes OC differentiation, also called osteoclastogenesis, via its receptor RANK leading to recruitment of TNF receptor-associated factor 6 (TRAF6) and, in turn, activation of multiple downstream targets including mitogen-activated protein (MAP) kinases, activator protein-1 (AP-1) and NF-κB[7]. RANKL-driven osteoclastogenesis is also dependent on the generation of a calcium signal through the activation of the immunoreceptor tyrosine-based activation

motifs (ITAMs) of DNAX-activation protein (DAP) 12 and Fc-receptor common γ subunit (FcRγ)[8]. This enables TRAF6-mediated and ITAM-mediated signals to interact cooperatively in transcriptional upregulation of nuclear factor of activated T cells c1 (NFATc1, the master transcription factor).

Recent reports indicate that intracellular ROS including superoxide anion and hydrogen peroxide ($H_2O_2$) play a second messenger role in the receptor-mediated signaling cascades in various type of cells[9–11]. Activation of RANK by RANKL during osteoclastogenesis results in the generation of ROS which reinforces activation of the RANKL-mediated signaling[12–15]. Consequently, RANKL-induced ROS further stimulate OC formation and bone resorption. Therefore, a delicate regulation of ROS levels is critical in maintaining bone homeostasis.

The antioxidant protein and a scavenger of ROS, DJ-1 also known as PARK7, is a 189-amino acid protein that is widely distributed in tissues[16, 17]. DJ-1 was originally identified as an oncogene product and aberrant expression of DJ-1 is associated with tumorigenesis in several cancer tissues[18]. It should be noted

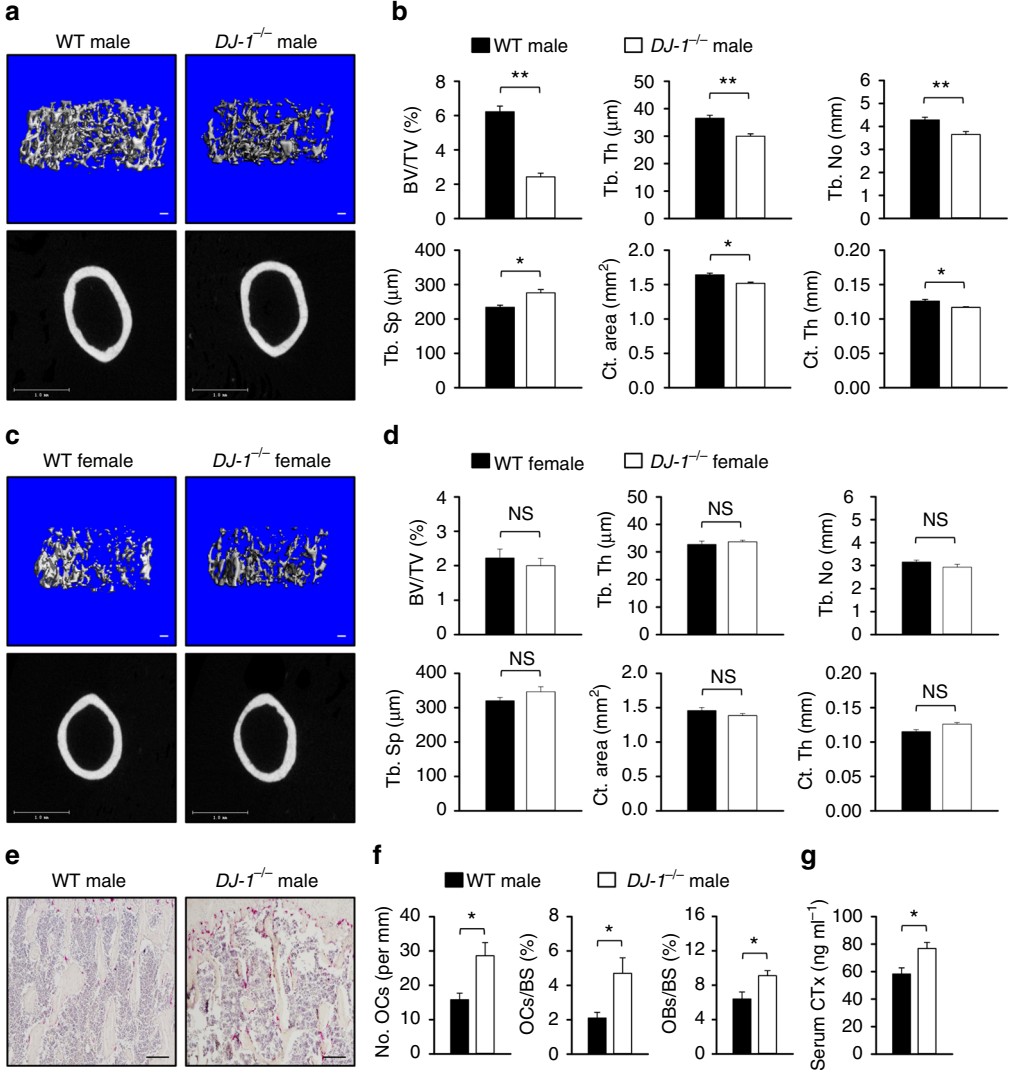

**Fig. 1** Decreased bone volume and increased OC number in $DJ\text{-}1^{-/-}$ mice. **a, c** Representative microCT images of trabecular (Tb) and cortical (Ct) bone from WT or $DJ\text{-}1^{-/-}$ male and female mice (both 10 weeks old, $n = 6$ per group, Scale bar; Tb 100 μm, Ct 1 mm). **b, d** Quantification of trabecular bone volume per tissue volume (BV/TV), Tb thickness (Th), number (No), spacing (Sp), Ct area, and Ct Th in femur tissues. **e** Representative images of femur from WT or $DJ\text{-}1^{-/-}$ male mice that were stained with TRAP. Magnification, ×100 and scale bar, 100 μm, $n = 6$ per group. **f** Histomorphometric analysis of OCs number, OCs surface and OBs surface in femur tissues. **g** The measurement of serum collagen degradation product (CTx) in WT or $DJ\text{-}1^{-/-}$ mice. **b, d, f, g** The data shown as the mean ± s.e.m. $n = 6$ per group. **$p < 0.01$, *$p < 0.05$, NS, not significant vs. WT by Student's $t$-test

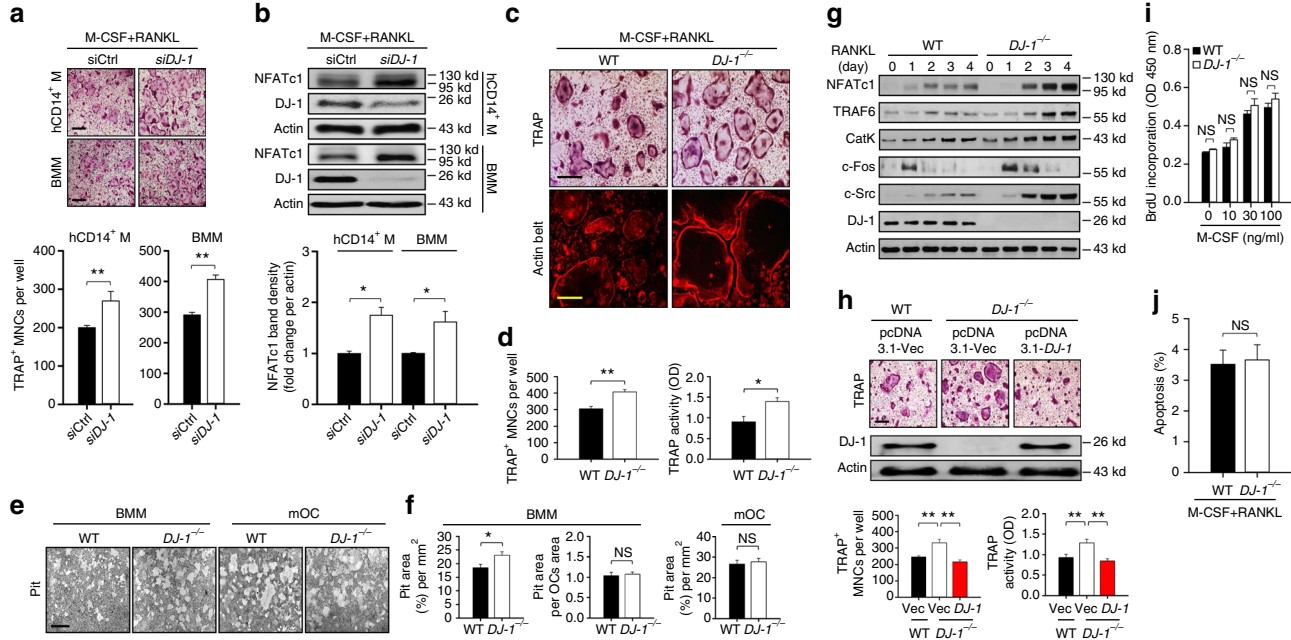

**Fig. 2** Deficiency of DJ-1 increases OC differentiation. **a** Representative images (upper panel) of TRAP+ multinucleated osteoclasts (OCs) in *DJ-1* siRNA transfected human CD14+ monocytes (hCD14+ M) or mouse BMMs. Magnification, ×100 and scale bar, 200 μm. The number of TRAP+ OCs (lower panel). **b** Representative immunoblot images (upper panel) for NFATc1 in human CD14+ monocytes or mouse BMMs stimulated by RANKL for 48 h. Band density of NFATc1 (lower panel). **c** Representative images for TRAP+ OCs (Magnification, ×100 and scale bar, 200 μm) and actin belt formation (rhodamine-phalloidin; Magnification, ×200 and scale bar, 100 μm) in WT or *DJ-1⁻/⁻* BMMs by RANKL. **d** The number of TRAP+ OCs and TRAP activity in WT or *DJ-1⁻/⁻* BMMs for the upper panel of **c**. **e** Representative images for bone resorption by OCs. Magnification, ×200 and scale bar, 100 μm. **f** Pit area (left), pit area per OCs area (middle) by BMMs and pit area by mOCs (right). **g** Representative immunoblot images for OC marker proteins in WT or *DJ-1⁻/⁻* BMMs. **h** Representative image for TRAP+ OCs (upper panel, Magnification, ×100 and scale bar, 200 μm). The number of TRAP+ OCs and TRAP activities were measured in WT or *DJ-1⁻/⁻* BMMs overexpressed with *DJ-1* plasmid or empty vector (lower panel). Representative images and data shown as the mean ± s.e.m. from three independent experiments in triplicate. **p < 0.01 by one-way ANOVA with post hoc Tukey's test. **i** Measurement of proliferation by BrdU incorporation in WT or *DJ-1⁻/⁻* BMMs for 3 days; n = 5. **j** Frequency of apoptotic cells in BMMs for 2 days. **a**, **b**, **d**, **f**, **i**, **j** The data shown as the mean ± s.e.m. from three independent experiments in triplicate. **p < 0.01, *p < 0.05, NS, not significant vs. WT by Student's *t*-test

that DJ-1 is also linked to an early onset of autosomal recessive Parkinson's disease (PD) via homozygous deletion and loss of function mutation of *DJ-1* gene[19]. According to several reports, DJ-1 scavenges ROS through self-oxidation of DJ-1 at cysteine residues 46, 53, and 106[20] and it also interacts with superoxide dismutase, glutathione peroxidase, and catalases to enhance their ability to remove ROS[21–23]. Other reports suggest that dysfunction of DJ-1 contributes to onset and severity of various diseases including Parkinson's disease, sclerosis, hypertension, obesity, and allergy as a result of oxidative stress[24–28]. However, to our knowledge, the role of DJ-1 in osteoclastogenesis remains unclear.

In this study, we demonstrate that DJ-1 plays a critical role in regulation of osteoclastogenesis in normal physiology, as well as bone-associated pathology. DJ-1 increases the activity of Src homology region 2 domain-containing phosphatase-1 (SHP-1) by decreasing intracellular ROS concentration during RANKL-stimulated osteoclastogenesis. The activated SHP-1 inhibits RANK-mediated signals and eventually inhibits osteoclastogenesis. Thus, defect in DJ-1 may result in impaired bone homeostasis.

## Results

**DJ-1 regulates bone mass and number of OC in mice.** The three-dimensional microstructural analysis was performed using high-resolution micro-computed tomography (μCT) to determine the bone mass of 10-week-old wild-type (WT) and DJ-1 knockout (KO) male and female mice. Interestingly, this analysis indicated that bone loss was observed only in male mice (Fig. 1), in which, DJ-1 deficiency caused decreases of other bone

morphological parameters, such as trabecular bone volume (BV/TV, by 57.6%), trabecular thickness (Tb. Th, by 16.4%), trabecular number (Tb. No, by 14.8%), cortical area (Ct. area, by 7.4%) and cortical thickness (Ct. Th, by 7.3%). In contrast, trabecular spacing increased (Tb. Sp, by 18.0%) in DJ-1KO mice (Fig. 1a, b). Histomorphometric analysis revealed significant increases in the number of OCs (by 81.0%), OC surface (by 123.8%), and osteoblast surface (by 42.1%) in DJ-1 KO mice compared to WT mice (Fig. 1e, f). The amount of secreted collagen type 1 fragments (CTx) also increased in the serum of DJ-1 KO mice in comparison to that of WT mice (Fig. 1g). These results suggest that DJ-1 plays a role in the regulation of OC differentiation in vivo.

**DJ-1 regulates osteoclastogenesis in vitro.** We next examined whether DJ-1 also regulated RANKL-mediated OC differentiation in vitro in both human and mouse OC progenitor cells in the presence of M-CSF. The multinuclear tartrate-resistant acid phosphatase (TRAP)+, a marker of differentiated OCs, increased significantly in human peripheral blood mononuclear cell (PBMC)-derived CD14+ monocytes, and in mouse bone marrow macrophages (BMMs) after knockdown of *DJ-1* using siRNAs against human *DJ-1* and murine *DJ-1*, respectively (Fig. 2a). Consistent with these results, *DJ-1* knockdown significantly enhanced the expression of NFATc1, the key transcription factor for OC differentiation, in RANKL stimulated human CD14+ and murine BMMs (Fig. 2b).

We verified that DJ-1 was critical for OC differentiation by using BMMs from DJ-1 KO mice. Although there was no

difference in RANK expression between WT and DJ-1 KO BMMs (Supplementary Fig. 1), the number of multinucleated TRAP⁺ OCs, TRAP activity and actin belt formation increased significantly in DJ-1 KO BMMs compared to WT controls (Fig. 2c, d). In addition, the formation of lacunae in osteologic disk and collagen type 1 fragments in culture was enhanced in DJ-1 KO BMMs (Fig. 2e, f, Supplementary Fig. 2). However, there was no significant difference in the activity of bone resorption in mature OCs from WT or from DJ-1 KO-derived BMMs (Fig. 2e, f). Expression of the archetypical RANKL signaling mediators such as NFATc1, TRAF6, Cathepsin K (CatK), c-Fos and c-Src also increased in DJ-1 KO BMMs in comparison to WT BMMs (Fig. 2g). Notably, the elevated numbers of multinuclear TRAP⁺ OCs and TRAP activity in DJ-1 KO BMMs reverted to the levels of WT cells following overexpression of DJ-1 in these cells (Fig. 2h). In addition, we found that the increased number of OCs in DJ-1 KO BMM cell cultures did not attribute to changes in cell proliferation or apoptosis in response to M-CSF and/or RANKL (Fig. 2i, j).

**DJ-1 deficiency leads to increases of RANK signals.** The interaction of RANKL with RANK leads to recruitment of TRAF6 by RANK, activation of Src kinase, and consequently, activation of the Akt, MAP kinase and the NF-κB pathway. ITAM-mediated co-stimulatory signals involving Syk, PLCγ2, and Gab2 are also essential for RANKL-mediated OC differentiation[29–35]. In this context, we examined which of these two pathways were regulated by DJ-1. The RANKL-stimulated activating phosphorylation of Akt, IkBα, and three typical MAP kinases (ERK1/2, JNK, and p38) increased substantially in DJ-1 KO BMMs compared to WT controls (Fig. 3a). Enhanced activation of ITAM-mediated co-stimulatory signaling cascades as demonstrated by increased phosphorylation of Syk, PLCγ2, Gab2 (Fig. 3b) and activation of Syk (Fig. 3c) was also apparent in DJ-1 KO BMMs. However, such enhancement due to DJ-1 deficiency was ablated by over-expression of *DJ-1* in DJ-1 KO BMMs (Fig. 3d), suggesting that DJ-1 negatively regulates RANKL-stimulated both TRAF6 and ITAM-mediated pathways in BMMs. To further refine these results, we next examined the effect of DJ-1 deficiency on the activation of the RANK-associated early signals, and found that the association of RANK with TRAF6, as well as that of FcRγ with Syk on stimulation with RANKL increased in DJ-1 KO BMMs compared to WT BMMs (Fig. 3e, f).

**DJ-1 controls ROS concentration in RANKL stimulated BMMs.** As DJ-1 is recognized as a ROS scavenging protein[21], we investigated the correlation between the expression of DJ-1 and the concetration of ROS during osteoclastogenesis in BMMs. We found that treatment with ROS scavenger 2,2,6,6-tetramethylpiperidinyloxy (TEMPO) abolished the stimulatory effect of DJ-1 deficiency on RANKL-activated signaling molecules (Fig. 4a). Although RANKL itself caused an increase in the concentrations of ROS significantly of greater extent in DJ-1 KO BMMs than in WT BMMs (Fig. 4b, c), the enhancement of ROS concentrations in DJ-1 KO BMMs was reversed by over-expression of *DJ-1* (Fig. 4d) or by treatment with TEMPO (Fig. 4e). Similar results were also observed in human CD14⁺ monocytes using *DJ-1* siRNAs (Fig. 4f).

In addition, it is reported that ROS stimulates the activation of RANK/RANKL-mediated signals associated with osteoclastogenesis[14, 15]. We found accordingly that $H_2O_2$ stimulated both TRAF6-mediated and ITAM-dependent signaling cascades in BMMs (Fig. 4g, h). Also, treatment with TEMPO suppressed the RANKL-mediated OC differentiation in DJ-1 KO BMMs (Fig. 4i, j). Collectively, these results suggest that the effect of DJ-1 on

osteoclastogenesis is associated with decrease in accumulation of ROS in BMMs.

**DJ-1 regulates TRAF6 and ITAM-mediated signals via SHP-1.** SHP-1, also known as tyrosine-protein phosphatase non-receptor type 6, is a member of the protein tyrosine phosphatase (PTP) family. SHP-1 suppresses the activation of both RANK/TRAF6-mediated signals[36] and ITAM-mediated signals[37–40] during osteoclastogenesis. Furthermore, the activity of SHP-1 is inhibited by ROS[41–43]. Therefore, we investigated whether DJ-1 regulated the activation of SHP-1 through its action on ROS in RANKL-stimulated BMMs. The tyrosine phosphorylation of SHP-1 by RANKL (Fig. 5a) and the activation of SHP-1 (Fig. 5b) were significantly suppressed in DJ-1 KO BMMs compared to WT controls. Activation of SHP-1 was also suppressed following treatment with $H_2O_2$ to WT BMMs (Fig. 5c). Consequently, we found that SHP-1 was associated with TRAF6 and with a typical co-stimulatory ITAM receptor FcRγ in RANKL-stimulated BMMs, and that these interactions were inhibited significantly in DJ-1 KO cells in comparison to WT controls (Fig. 5d, e). As observed in DJ-1-deficient cells, $H_2O_2$ similarly reduced the association of SHP-1 with TRAF6 and with FcRγ in RANKL-stimulated WT BMMs (Fig. 5f–i). These results support the idea that DJ-1 promotes RANKL-induced association of SHP-1 with TRAF6 and with FcRγ by inactivating of ROS.

**DJ-1 deficiency increases ROS to stimulate osteoclastogenesis in vivo.** We tested to verify that the effect of DJ-1 deficiency was associated with the production of ROS in vivo. ROS production increased significantly in calvaria from DJ-1 KO mice after treatment with RANKL (Fig. 6a, b) or with lipopolysaccharide (Supplementary Fig. 3a, b) in comparison to the WT controls. Notably, DJ-1 deficiency resulted in a significant increase of osteoclastogenesis in calvarial tissues of mice injected with RANKL (Fig. 6c–e) or with LPS (Supplementary Fig. 3c, d), suggesting that the regulation of ROS by DJ-1 is also closely associated with osteoclastogenesis in vivo.

**DJ-1 deficiency exacerbates collagen-induced arthritis.** OCs are the only bone-resorbing cells that undergo bone destruction in RA[44]. Therefore, we investigated whether DJ-1 has a role in osteoclastogenesis during collagen-induced arthritis (CIA), a typical OC-associated pathology, in mice. We first measured the effect of CIA on ROS levels, which revealed that ROS levels were enhanced significantly in the serum of CIA-induced mice (35.2 ± 3.4 μM) in comparison to control mice (19.4 ± 2.5 μM) (Fig. 7a). We then examined the effect of DJ-1 on the severity of arthritis in CIA mice, which showed that the severity and the frequency of onset of arthritis in this model increased significantly in DJ-1 KO CIA mice (Fig. 7b–d). In addition, evaluation of the effects of DJ-1 on synovial inflammation and bone destruction in joint tissues revealed that immune cell infiltration, cartilage destruction, and bone erosion increased significantly in DJ-1 KO mice (Fig. 7e upper panel, f). This included a notable increase in the number of TRAP⁺ OCs in ankle joint tissues of DJ-1 KO mice as well (Fig. 7e lower panel, g). There was also a corresponding increase in expressions of typical OC markers in the ankle tissues of DJ-1 KO CIA mice including TRAP, Cathepsin K, calcitonin receptor, and OC-associated immunoglobulin-like receptor (Fig. 7h, i). These results suggest that intracellular DJ-1 in OC-precursor cells regulate, at least partially, osteoclastogenesis in the progression of arthritis in CIA mice.

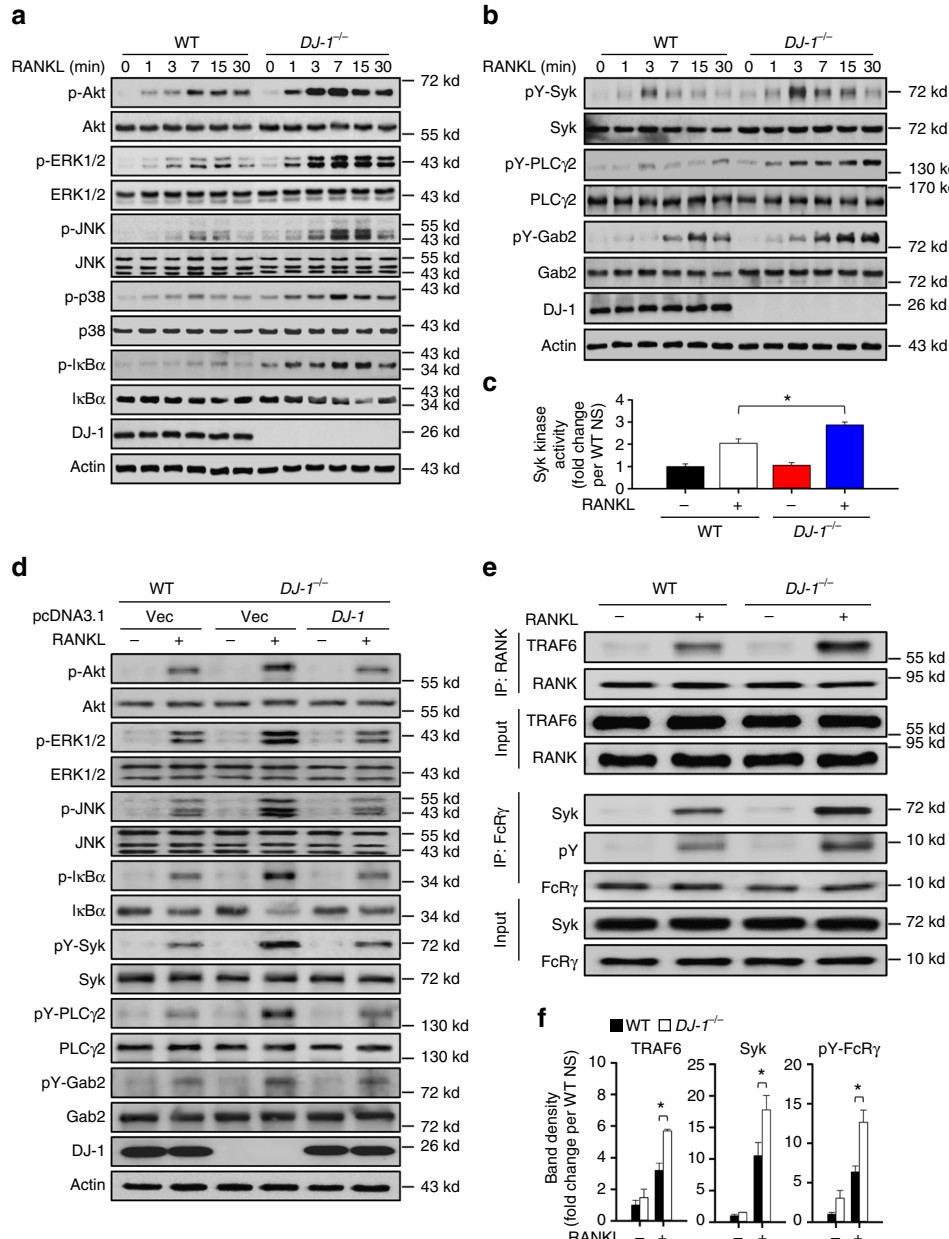

**Fig. 3** DJ-1 is critical to TRAF6 and ITAM-mediated signals. **a**, **b** Representative immunoblot images for RANKL-stimulated signals in WT or *DJ-1*⁻/⁻ BMMs. **c** Syk kinase activity in WT or *DJ-1*⁻/⁻ BMMs. *$p < 0.05$ vs. RANKL stimulated WT BMMs by Student's *t*-test. **d** Representative immunoblot images for RANKL-stimulated signaling proteins in WT, or *DJ-1*⁻/⁻ BMMs with or without overexpression of DJ-1 plasmid. **e** Representative images for co-immunoprecipitation of TRAF6 and RANK or FcRγ and Syk in WT or *DJ-1*⁻/⁻ BMMs. **f** Band density of the associations in **e**. *$p < 0.05$ vs. RANKL stimulated WT BMMs by Student's *t*-test. Representative images **a**, **b**, **d**, **e** and data **c**, **f** shown as the mean ± s.e.m. from three independent experiments in triplicate

## Discussion

DJ-1 is recognized as an antioxidant protein capable of destroying ROS[21–28] and there is accumulating evidence that it also plays a functional role in cell homeostasis. The mutation of DJ-1 is closely associated with familial Parkinson's disease, as well as with tumorigenesis and male fertility[18, 45]. In addition, there is evidence that DJ-1 regulates essential signaling molecules in several types of cells[28, 46–49]. Others have also reported that soluble recombinant DJ-1 facilitates osteoblast differentiation in human mesenchymal stem cells and angiogenesis in endothelial cells when these cells are stimulated by fibroblast growth factor receptor-1[46]. However, the role and mechanism of intracellular DJ-1 in the differentiation of OC, a cell type that plays an essential role in bone metabolism and RA pathology, remain unknown.

Although OCs, along with osteoblasts, are closely associated with bone homeostasis in normal physiology, they also play an adverse role in the process of bone destruction in RA, osteoporosis, and Paget disease[3–5]. In this study, we assessed the role of DJ-1 in bone metabolism under both normal and pathological states in a series of experiments. We found that while trabecular and cortical bone volumes decreased in DJ-1 KO male mice, in comparison to WT male mice (Fig. 1), the population of OCs, and to a lesser extent, of osteoblasts increased in femurs (Fig. 1). DJ-1 deficiency also resulted in an increase in osteoclastogenesis in vitro cultured human CD14⁺ monocytes and BMMs with knockdown of DJ-1 by using siRNAs (Fig. 2a), as well as in BMMs from DJ-1 KO mice (Fig. 2c, d). As DJ-1 deficiency did not alter the extent of proliferation or apoptosis of BMMs

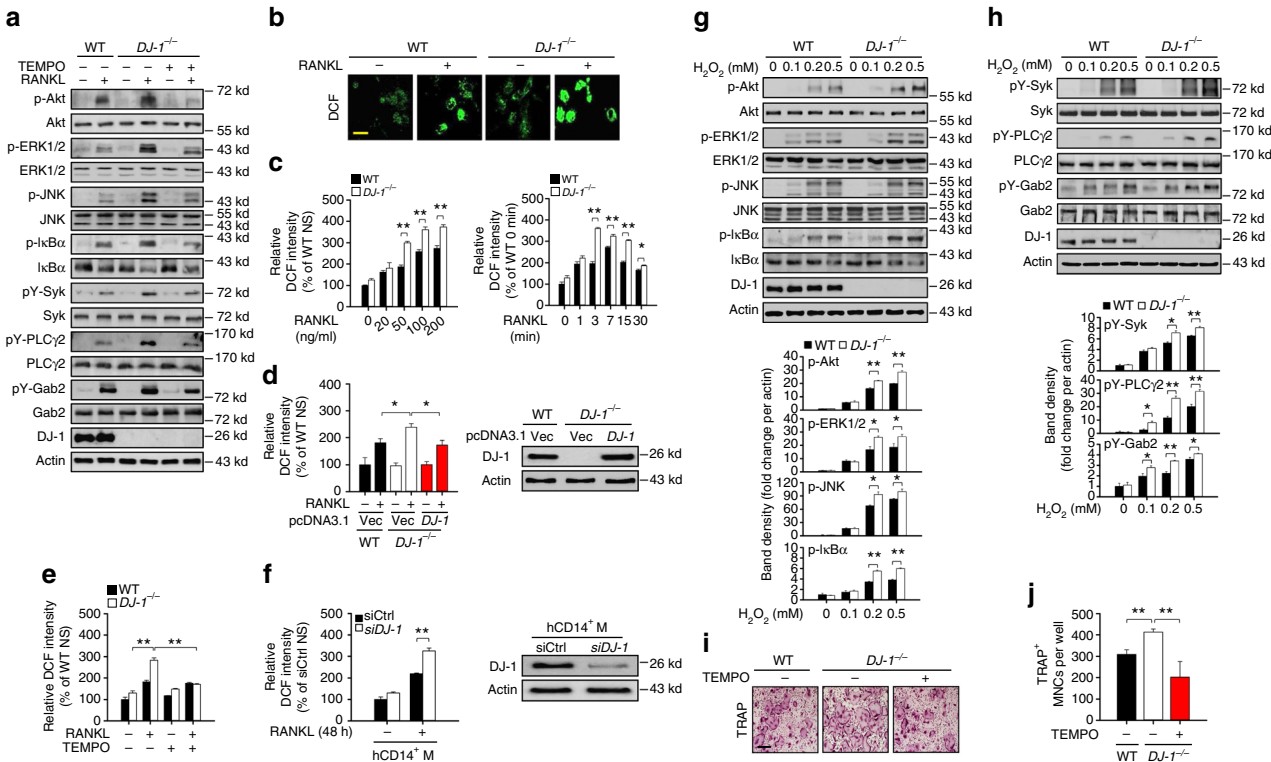

**Fig. 4** DJ-1 controls the RANK signals via suppression of ROS. **a** Representative immunoblot images for phosphorylated forms of signaling proteins stimulated by RANKL with or without TEMPO (30 μmol L$^{-1}$) in WT or $DJ-1^{-/-}$ BMMs. **b** Representative images for DCF fluorescence in WT or $DJ-1^{-/-}$ BMMs stimulated with 100 ng ml$^{-1}$ RANKL for 10 min. Magnification, ×400 and scale bar, 50 μm. **c** Relative DCF intensity in BMMs under the indicated conditions. **d** Relative DCF intensity shown as the mean ± s.e.m. (left) and representative images (right) in WT or $DJ-1^{-/-}$ BMMs with the transfection of $DJ-1$ plasmid or empty vector from three independent experiments in triplicate. **p < 0.01 by one-way ANOVA with post hoc Tukey's test. **e** Relative DCF intensity in WT or $DJ-1^{-/-}$ BMMs incubated with or without 30 μmol L$^{-1}$ TEMPO. **f** Relative DCF intensity in human CD14$^+$ monocytes stimulated by RANKL for 48 h after the transfections of control siRNAs (siCtrl) or $DJ-1$-targeted siRNAs (siDJ-1). **g, h** Representative immunoblot images (upper panel) and relative band densities (lower panel) for essential signaling proteins for OC differentiation in BMMs stimulated by H$_2$O$_2$ for 15 min **g** or 5 min **h**. **i** Representative images (magnification, ×100 and scale bar, 200 μm) and **j** number of TRAP$^+$ OCs in RANKL-induced WT and $DJ-1^{-/-}$ BMMs with or without TEMPO (30 μmol L$^{-1}$ from three independent experiments in triplicate. **p < 0.01 by one-way ANOVA with post hoc Tukey's test. Representative images **a, b, f–i** or the mean ± s.e.m. **c, e–h** from three independent experiments in triplicate are shown. **p < 0.01, *p < 0.05. by Student's t-test

(Fig. 2i, j), DJ-1 appeared to directly regulate osteoclastogenesis and consequently, bone metabolism. On the other hand, bone surface of OB also increased in DJ-1 KO mice (Fig. 1f). However, unexpectedly, OB differentiation in vitro was slightly reduced in DJ-1 KO cells, in comparison to WT cells (Supplementary Fig. 4). These results suggest that the increase of OBs in vivo is mediated by a secondary feedback mechanism of bone loss in DJ-1 KO mice and not by the cell autonomous activity.

Interestingly, we found that there was a sex-specific response to the loss of DJ-1 in terms of bone phenotype (Fig. 1). This was not surprising, as we and others have previously observed in several gene-inactivated KO mice model that those animals exhibited a sex-specific bone phenotype at the same age examined in this study[50–52]. While male mice showed changes in the trabecular and cortical bone, female mice showed none (Fig. 1), demonstrating a sex-specific response for which the mechanism is currently unclear. The changes in the bone phenotype that we observed in male mice may be stage-specific for male and female mice show difference in their development of bone growth as well as their point of time reaching their peak bone mass. It may also be possible that female mice would have eventually achieved a similar decrease in trabecular bone mass at a later point of time than that shown in male mice at the age of 6–10 weeks, or that DJ-1 may not be a negative regulator of osteoclastogenesis in female mice. These results warrant further investigation on sexual dimorphism on bone phenotype attributed by DJ-1.

In humans, the prevalence of osteoporosis and RA are overwhelmingly higher in women than in men of age similar to that of menopausal women[3, 53, 54]. For this reason, many studies have been conducted in female patients with bone-associated pathology. Although osteoporosis and RA in men are also recently increasing, information on the risk factors and characteristics of bone-associated pathology in men is relatively limited in comparison to that of women[55, 56]. Therefore, clinical research of the function of DJ-1 in various bone-associated pathology, especially in male patients, will be important in the future.

Both TRAF6- and ITAM-mediated signaling cascades are critical for RANKL-stimulated formation of OCs from precursor monocytes[29, 30]. Interestingly, we found that both cascades were enhanced in DJ-1 KO BMMs (Fig. 3a–c), which suggests that DJ-1 plays the role through a common mechanism or step of both RANK-signaling cascades. The recruitments of TRAF6 by RANK and Syk by FcRγ are important in activating the two downstream cascades[57, 58]. Notably, we observed that both recruitments were enhanced in DJ-1 KO BMMs in comparison to WT controls (Fig. 3e, f). These results suggest that DJ-1 acts at an early stage of both pathways namely the recruitment of TRAF6 by RANK and of Syk by FcRγ.

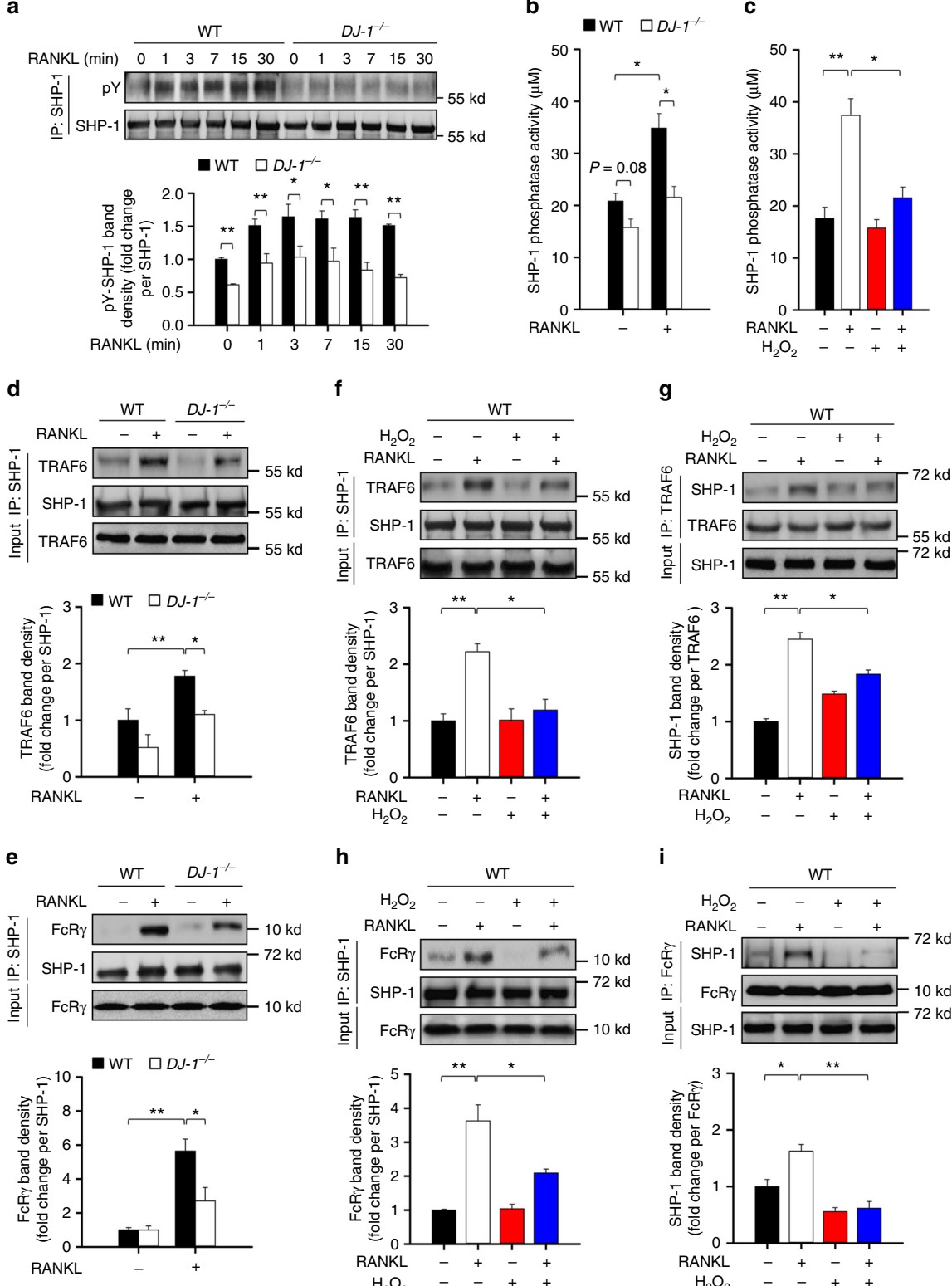

**Fig. 5** DJ-1 deficiency inhibits SHP-1 in BMMs. **a** Representative immunoblot images (upper panel) and densities (lower panel) for phosphorylated forms of SHP-1 after immunoprecipitation of SHP-1 shown as the mean ± s.e.m. from three independent experiments in triplicate. **p < 0.01, *p < 0.05 by Student's t-test. **b**, **c** Activity of immunoprecipitated SHP-1 in WT or $DJ-1^{-/-}$ BMMs stimulated by RANKL for 7 min with or without $H_2O_2$ (100 μM) as indicated. **d**, **e** Representative immunoblot images and band densities for co-immunoprecipitations of SHP-1 with TRAF6 or FcRγ by RANKL in WT or $DJ-1^{-/-}$ BMMs. **f**–**i** Representative immunoblot images and band densities for co-immunoprecipitations of SHP-1 with TRAF6 or FcRγ with or without $H_2O_2$ (100 μM) in BMMs. All representative images **a**, **d**–**i** and the mean ± s.e.m. **a**–**i** are taken from three independent experiments in triplicate. **p < 0.01, *p < 0.05 by one-way ANOVA with post hoc Tukey's test **b**–**i**

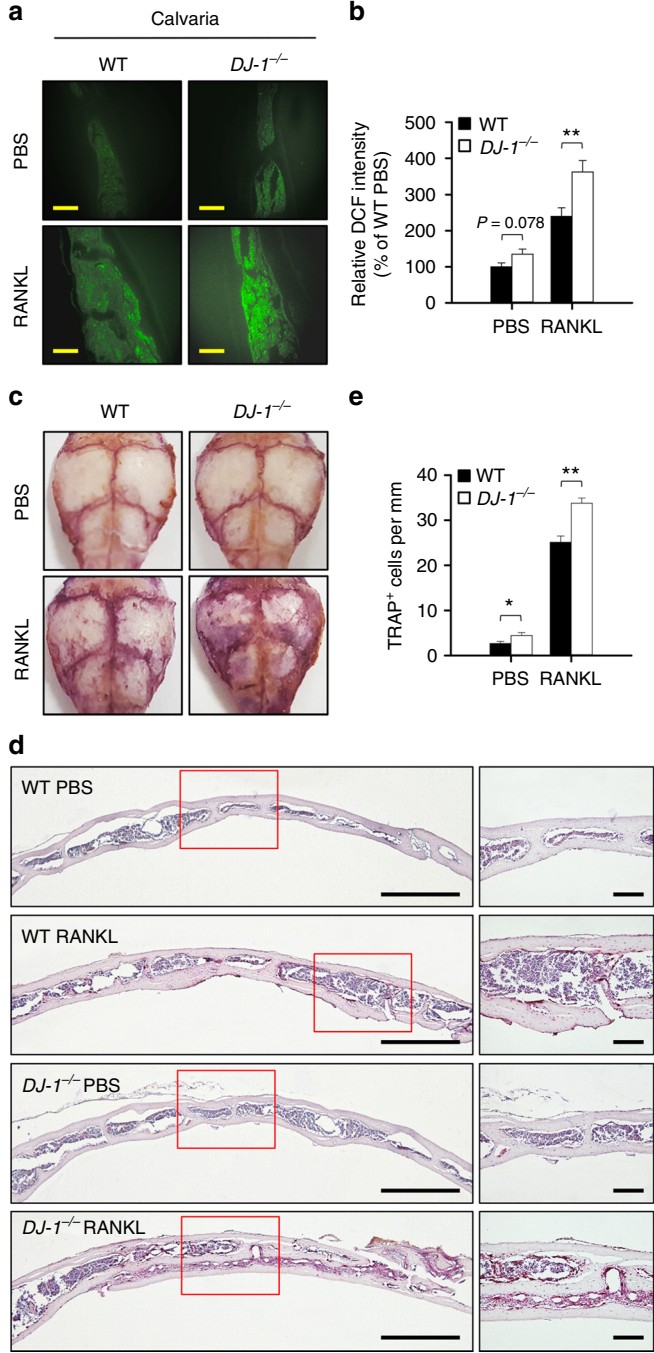

**Fig. 6** DJ-1 deficiency increases ROS and OC number in mice. **a** Representative images for DCF fluorescence and **b** relative DCF intensities in RANKL-stimulated calvaria in WT and *DJ-1⁻/⁻* mice. Scale bar, 100 μm. **c**–**e** Representative images for TRAP-stained whole **c** and sectioned **d** calvaria tissues in RANKL-treated WT and *DJ-1⁻/⁻* mice. Scale bar, 500 μm (left), 100 μm (right). **e** Number of TRAP⁺ cells in cavaria tissues in RANKL-injected WT and *DJ-1⁻/⁻* mice. Representative images **a**, **c**, **d** or the mean ± s.e.m. **b**, **e** are shown. $n = 6$ per group. $**p < 0.01$, $*p < 0.05$ by Student's *t*-test

In contrast to DJ-1, ROS is recognized to play a positive role in the regulation of RANKL-induced osteoclastogenesis in OC precursor cells[11, 14, 59], in the activation of multiple signaling proteins, including PI3K, Akt, MAP kinases, and NF-κB, in various cell types (reviewed in refs. [60–62]). RANKL-induced ROS production is also essential for OC differentiation via TRAF6,

Rac1, NADPH oxidase (Nox) 1 and MAP kinases[15]. Due to the known roles of ROS in osteoclastogenesis and cell signaling, we investigated whether DJ-1 could alter RANK-mediated signals by decreasing ROS amounts during osteoclastogenesis. We observed that ROS amounts increased with RANKL stimulation of BMMs and was enhanced further in DJ-1 KO BMMs, but were inhibited by treatment with ROS scavenger molecule TEMPO (Fig. 4a, e), indicating that the osteoclastogenic effect of DJ-1 is associated with the amount of ROS in BMMs. We also observed that high concentrations of ROS (i.e., $H_2O_2$) activated most of the essential signaling molecules involved in both the TRAF6 and ITAM-mediated co-stimulatory pathways during osteoclastogenesis (Fig. 4g, h). Consistent with the above results, treatment with TEMPO suppressed RANKL-induced osteoclastogenesis in DJ-1 KO BMMs (Fig. 4i, j). Collectively, these results support the idea that an increase in ROS enhances RANK-mediated TRAF6 and ITAM signals during osteoclastogenesis, while DJ-1 counteracts such effect of ROS.

The protein tyrosine phosphatase SHP-1 is a well-known negative regulator of signaling pathways in various cells[63], among which, binding of SHP-1 with TRAF6 or with ITAM-containing receptors in particular leads to de-ubiquitination of TRAF6 or dephosphorylation of phosphorylated ITAMs, respectively, in RANKL-activated BMMs[36–40]. SHP-1 is also reported to be directly associated with TRAF6 in RANKL-stimulated BMMs with subsequent inhibition of downstream signaling events such as MAP kinases and NF-κB[36]. Meanwhile, previous pertinent studies suggest that ROS inhibits the activity of SHP-1[41–43]. Therefore, we hypothesized that the effect of DJ-1 on TRAF6 and ITAM-mediated signals is mediated through its activation of SHP-1 by regulating ROS amounts. In this study, we observed that DJ-1 was essential for the activation of SHP-1 in RANKL-stimulated BMMs (Fig. 5a, b), while that the activation of SHP-1 was inhibited by treatment of $H_2O_2$ (Fig. 5c), indicating that intracellular ROS, in contrast to DJ-1, negatively regulates the activation of SHP-1 in BMMs. We also showed that RANKL-mediated association of SHP-1 with TRAF6 or with FcRγ was significantly suppressed in DJ-1 KO BMMs and in WT BMMs treated with $H_2O_2$ (Fig. 5d–i). These results again suggest that DJ-1 regulates the osteoclastogenesis through its activation of SHP-1 by decreasing ROS amounts.

DJ-1 is involved in various intracellular signaling pathways, including those associated with cancer and Parkinson's disease[49]. Intracellular DJ-1 modulates protein activity by regulating ROS concentration or by directly interacting with other signaling molecules. DJ-1 physically binds to PTEN, MEKK1, p53, Daxx, and Nrf2[49] and it also regulates the activity of AMP-activated protein kinase, SHP-1, and SHP-2 by modulating intracellular ROS concentration[28, 64]. Recently, it was reported that DJ-1 functions as a scaffold protein for STAT1 and SHP-1 interactions in the brain microglia and astrocytes[65]. In our experiment, SHP-1, which plays a critical role in osteoclastogenesis[36], was inhibited in cells treated with ROS as well as in DJ-1 KO cells (Fig. 4b, c). However, we could not observe any association between DJ-1 and SHP-1 by RANKL stimulation. Our results suggest that DJ-1 regulates the activity of SHP-1 by adjusting intracellular ROS during osteoclastogenesis.

It is generally accepted that OC is the only cell that undergoes bone resorption in arthritis and in bone metabolism[3]. Therefore, it is interesting to determine whether DJ-1 plays an important role in the differentiation of OC in vivo. In our study, the number of osteoclast was increased in DJ-1 KO arthritic mice, compared to WT arthritic mice (Fig. 7g). Likewise, the amount of mature OC-specific proteins such as TRAP, cathepsin K, calcitonin receptor, and osteoclast-associated immunoglobulin-like receptor was also increased in the joint tissues of DJ-1 KO arthritic animal

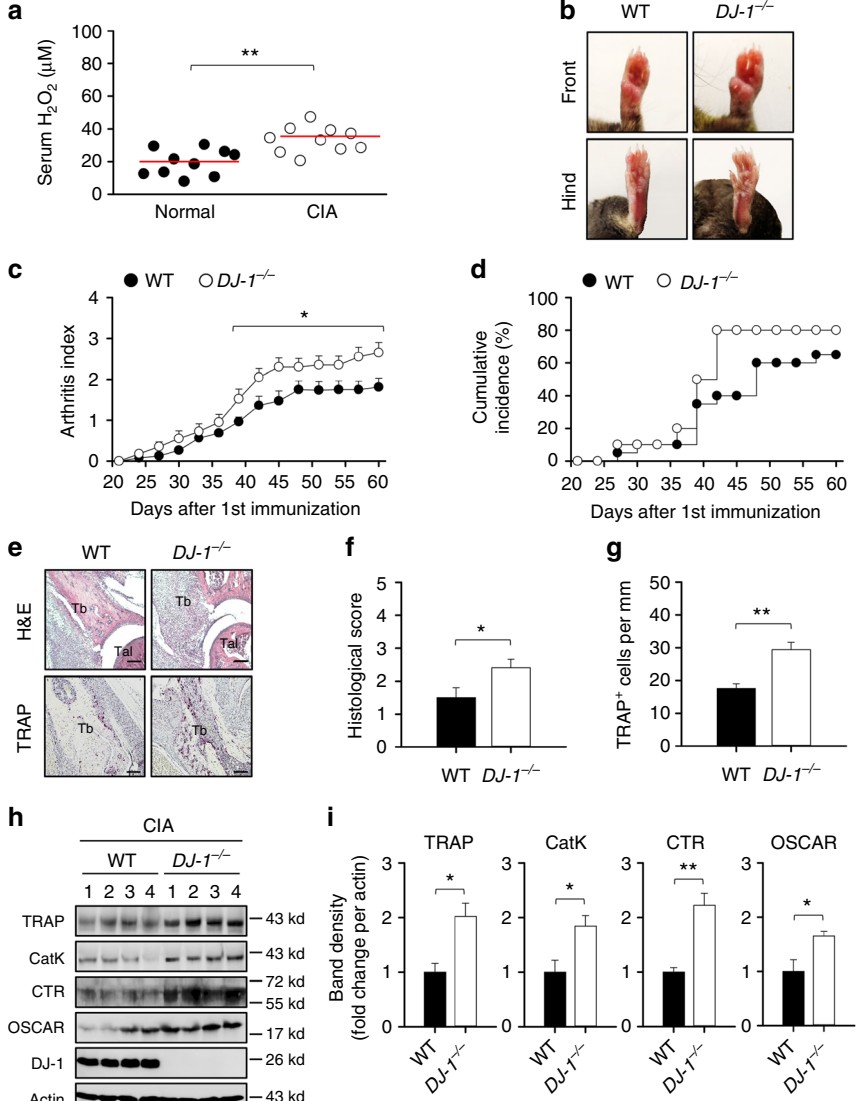

**Fig. 7** DJ-1 deficiency exacerbates collagen-induced arthritis in mice. **a** Sera from normal or CIA-induced C57BL/6 mice was measured by using ELISA ($n = 10$). **b** Representative images of front (upper) and hind (lower) paws of CIA-induced WT and $DJ-1^{-/-}$ mice. **c** Arthritis index values and **d** cumulative incidence are shown ($n = 20$). **p < 0.01 vs. corresponding values of WT mice by Mann−Whitney test for **c**. **e**–**g** Paraffin sections were prepared from CIA mouse ankle tissues stained with H&E (upper) and TRAP (lower) from experiments as in **b** (Tb: tibia, Tal: talus). Representative H&E and TRAP stained images **e**, histological scores **f**, and TRAP+ cell count **g** are shown. Scale bar, 50 μm. Representative immunoblot images **h** and band densities **i** for OC marker proteins of ankle tissues from experiments as in **b**; $n = 4$. Representative images **b**, **e**, **h** or values **c**, **f**, **g**, **i** are represented as the mean ± s.e.m. **p < 0.01, *p < 0.05 by Student's $t$-test **a**, **f**, **g**, **i**

models (Fig. 7h, i). These results suggest that DJ-1 in OC-precursor cells contributes, at least partially, to osteoclastogenesis in arthritis. However, the increase of OCs in DJ-1 KO CIA mice does not preclude the non-specific global effect by DJ-1 deficiency. We therefore compared the number of OCs in calvaria after injecting RANKL, which does not cause inflammation, into WT and DJ-1 KO mice. The number of OCs in calvaria of RANKL-injected DJ-1 KO mice significantly increased compared to WT mice (Fig. 6). Collectively, our results suggest that intracellular DJ-1 of OC-precursor cells is also critical for the osteoclastogenesis in vivo.

In summary, our results demonstrate for the first time that an ROS scavenging protein DJ-1 negatively regulates RANKL-stimulated signals during osteoclastogenesis via modulating SHP-1 (Fig. 8). Therefore, DJ-1 is a potential therapeutic target for osteoporosis and other bone-associated pathology such as RA.

## Methods

**Mice.** Wild-type (WT; $DJ-1^{+/+}$) and DJ-1-deficient (B6.Cg-$Park7^{tm1Shn}$/J; $DJ-1^{-/-}$) mice with the C57BL/6 background were purchased from The Jackson Laboratory (Bar Harbor, Maine, USA). All the mice were 6–10-weeks old and maintained in a specific pathogen-free housing facility of Konkuk University (Seoul, Korea) with a sterilized diet and autoclaved water, unless indicated otherwise. The Male mice were used for all in vitro experiments. All experiments were approved by the Institutional Animal Care and Use Committee (IACUC) at Konkuk University.

**Bone density and histomorphometric analysis.** Femurs from WT and $DJ-1^{-/-}$ male and female mice were fixed in 70% ethanol at 4 °C for microCT analysis. Trabecular and cortical morphometry within the metaphyseal region of the distal femur was quantified by using microCT (μCT40; Scanco Medical AG, Bassersdorf, Switzerland). Three-dimensional images were reconstructed by standard convolution back-projection algorithms with Shepp and Logan filtering and rendered at a discrete density of 578, 704 voxels (mm³)⁻¹ (isometric 12-mm voxels). Threshold segmentation of bone from marrow and soft tissue was performed in conjunction with a constrained Gaussian filter to reduce noise. Volumetric regions for trabecular analysis were selected within the endosteal borders to include secondary spongiosa of the femur (1 mm from the growth plate and extending 1 mm

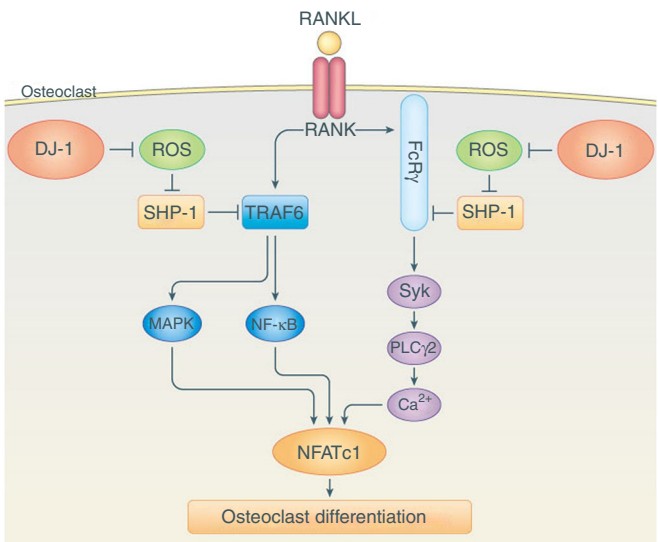

**Fig. 8** A proposed scheme for DJ-1 regulation of osteoclastogenesis. The binding of RANKL to RANK activates both TRAF6- and co-stimulatory ITAM-mediated signaling pathways. Both pathways are essential for full amplification of NFATc1 and as a consequence osteoclastogenesis. The activation of RANK also results in production of ROS which inhibits SHP-1. Our results indicate that the presence of DJ-1 alleviates this inhibition of SHP-1 by reducing the levels of ROS. As SHP-1 negatively regulates both TRAF6- and ITAM-mediated co-stimulatory signaling pathways, intracellular DJ-1 facilitates activation of both pathways and osteoclastogenesis by RANKL in vitro and in vivo

proximally). Trabecular morphometry was determined by measuring the bone volume fraction (BV/TV), trabecular thickness (Tb. Th), trabecular number (Tb. No), trabecular spacing (Tb. Sp), cortical area (Ct. area), and cortical thickness. Bone histomorphometric analysis was performed in a blinded, non-biased manner by using a computerized semi-automated system (Osteomeasure, Nashville, TN, USA) with light microscopy. OCs were quantified in paraffin embedded tissues that were stained for TRAP. OCs were identified as TRAP+ multinucleated cells adjacent to bone. The measurement terminology and units used for microCT and histomorphometric analyses were those recommended by the Nomenclature Committee of the American Society for Bone and Mineral Research[66, 67]. Briefly, all measurements were confined to the secondary spongiosa and restricted to an area between 400 and 2000 μm distal to the growth plate metaphyseal junction of the distal femur.

**RANKL-induced bone destruction in mouse calvaria.** Seven-week-old male mice were injected subcutaneously above the calvariae with 1 mg kg$^{-1}$ of RANKL (PeproTech Inc, Rock Hill, NJ, USA) in PBS daily for 5 days. After 5 days, whole calvariae were removed, fixed in 4% paraformaldehyde, and stained for TRAP using a leukocyte acid phosphatase kit (Sigma-Aldrich, St. Louis, MO, USA). To determine TRAP+ cell count, calvarial tissues were fixed in 4% paraformaldehyde in PBS for 6 days at 4 °C, decalcified in 14% EDTA and 9% NH$_4$OH solution (28%) in distilled water (pH 7.2) for 7 days at room temperature, and embedded in paraffin. Serial 5-μm paraffin sections were stained for TRAP with the leukocyte acid phosphatase kit and counterstained with hematoxylin.

**Induction and evaluation of collagen-induced arthritis.** C57BL/6 mice (6-week-old, male) were injected intradermally in the tail with 200 μg of chick type II collagen. The collagens were emulsified in complete Freund's adjuvant containing heat-killed *Mycobacterium tuberculosis* (Chondrex, Redmond, WA, USA). Twenty-one days after the first injection of collagen, the mice received booster injection of 100 μg of chick type II collagen in incomplete Freund's adjuvant (Chondrex). The arthritis severity was evaluated in average combined score of all four paws of the mice by three independent observers. The severity was scored as follows: 0, normal; 1, erythema and mild swelling; 2, erythema and slight swelling extending from the ankle to metatarsal joints; 3, erythema and moderate swelling; 4, Erythema and severe swelling encompass the ankle, foot and digits[68, 69]. To evaluate arthritis incidence, mice with an average severity score of 1 or greater were judged to be arthritis mice. All mice were killed 60 days after the start of the experiment and the following experiments were conducted. For histopathological analysis and determination of TRAP+ cell numbers, the decalcified and paraffin-embedded ankle tissue sections were stained with hematoxylin and eosin (H&E) or TRAP. Three

pathologists, who were unaware of the source of the tissues, independently evaluated each section on a 5-point scale using the following score: 0, normal; 1, minor destruction of the cartilage surface and infiltration of inflammatory cells; 2, mild hyperplasia of the synovial lining layer and slight cartilage destruction; 3, moderate cartilage destruction and pannus formation; 4, severe cartilage destruction and bone erosion; and 5, severe infiltration of inflammatory cells, bone erosion and severe destruction of cartilage. TRAP+ cell numbers were determined as described above for histomorphometric analysis.

**Osteoclast differentiation.** Human peripheral blood mononuclear cells (PBMCs) were isolated from blood of healthy volunteers by centrifugation over Ficoll/Paque Plus (GE healthcare, Uppsala, Sweden), and CD14+ monocytes were isolated from the PBMCs as previously reported[70]. Briefly, human PBMCs (1 × 10$^7$ cells) were incubated with human anti-CD14 mAb conjugated micro-beads (Miltenyi Biotec, Bergisch Gladbach, Germany) for 15 min at 4 °C. The cells were suspended in a separation buffer (1 × PBS, pH 7.2, 0.5% BSA and 2 mM EDTA) and then separated CD14+ monocytes on LS columns (Miltenyi Biotech). The cell preparations contained >95% CD14+ monocytes as determined by using the FACSCalibur flow cytometer (BD Biosciences, San Jose, CA, USA). Informed consents were obtained from all volunteers and the protocol was approved prior to the study from the Institutional Review Board of Konkuk University Hospital Clinical Trial Center (IRB No. KUH1010836; Seoul, Korea). Human CD14+ monocytes (1 × 10$^5$ cells per well in 96-well cluster plate) were incubated in α-MEM supplemented with 10% FBS, 2 mM L-glutamine, 100 U ml$^{-1}$ penicillin–streptomycin, 10 ng ml$^{-1}$ of M-CSF (PeproTech Inc.), 20 ng ml$^{-1}$ of RANKL (PeproTech Inc.) for 14 days. Whole bone marrow cells were isolated from long bones (femurs and tibias) of mice. The cells were cultured in α-MEM supplemented with 10% FBS for 24 h. Non-adherent cells were then cultured in α-MEM containing 10% FBS, 2 mM L-glutamine, 100 U ml$^{-1}$ penicillin-streptomycin, and 30 ng ml$^{-1}$ M-CSF (PeproTech Inc.) for 3 days. After 3 days, the attached cells were used as bone marrow-derived macrophages (BMMs). For determination of OC differentiation, BMMs (1 × 10$^4$ cells per well in 96-well cluster plastic plate were incubated in α-MEM containing with 10% FBS, 2 mM L-glutamine, 100 U ml$^{-1}$ penicillin–streptomycin, 30 ng ml$^{-1}$ of M-CSF, and 100 ng ml$^{-1}$ of RANKL (PeproTech Inc.) for 4 days. To identify OCs, actin belt formation was determined after fixing cells in 4% paraformaldehyde, permeabilized in 0.1% Triton X-100, and immunostained with rhodamine-phalloidin (Invitrogen, Carlsbad, CA, USA). TRAP activity was analyzed by using a p-nitrophenyl phosphate assay kit (Sigma-Aldrich), and then the cells were stained for TRAP with a Leukocyte acid phosphatase kit (Sigma-Aldrich). In this process, TRAP+ multinucleated cells that contained more than three nuclei were counted as OCs.

**Differentiation of osteoblastic stromal cells.** Whole bone marrow cells were isolated from long bones (femurs and tibias) of male mice. The cells were cultured in α-MEM containing with 50 μg ml$^{-1}$ ascorbic acid (Sigma-Aldrich) and 10 mM β-glycerophosphate (Sigma-Aldrich) for 21 days. The medium was changed twice a week. Gene expression level of alkaline phosphatase and osteocalcin was performed by reverse transcriptase-polymerase chain reaction (RT-PCR). The primers were used: mouse *ALP* forward 5′-CGCACGCGATGCAACACCAC-3′, reverse 5′-ACTGCATGTCCCCGGGCTCA-3′; mouse osteocalcin forward 5′-CCACA-CAGCAGCTTGGTGCA-3′, reverse 5′-CCCGGAGAGCAGCCAAAGCC-3′; mouse *GAPDH* forward 5′-TGACGTGCCGCCTGGAGAAA-3′, reverse 5′-AGTGTAGCCCAAGATGCCCTTCAG-3′. The mineralization was assessed by using an alizarin red incorporation measurement at 405 nm absorbance.

**In vitro proliferation and apoptosis assays.** Proliferation was assessed in BMMs grown in M-CSF (0–100 ng ml$^{-1}$) alone for 3 days by use of the Cell proliferation BrdU kit (Roche Diagnostics, Mannheim, Germany) and measurement of absorbance at 450 nm according to the manufacturer's instructions. After BMMs were cultured with M-CSF and RANKL for 2 days, apoptotic cells were detected by Annexin V assay kit (Phoenix Flow Systems, San Diego, CA, USA). Cells were analyzed by using the FACSCalibur flow cytometer (BD Biosciences).

**Transfection of DNA plasmid or small interfering RNAs.** Human PBMC-derived CD14+ monocytes and mouse BMMs (1 × 10$^6$ cells) were transfected with 30 pmoles of each species specific small interfering (si) RNAs against *DJ-1* for knockdown of DJ-1 or control of siRNAs. siGENOME ON-TARGETplus SMARTpool targeting *DJ-1* with four siRNA duplexes, or ON-TARGETplus siCONTROL non-targeting pool for control were purchased from Dharmacon (Lafayette, CO, USA). Mouse BMMs (1 × 10$^6$ cells) were transfected with 1 μg of pcDNA3.1-*DJ-1* or pcDNA3.1 vector plasmids for overexpression by electroporation with the Amaxa nucleofector (Lonza Cologne AG, Cologne, Germany). Immediately after electroporation, the cells were suspended in culture medium and transferred to culture plates. The program used for transfection with the Amaxa nucleofector was program No. Y—001 with Human Monocyte or Mouse Macrophage Nucleofector kit. Successful gene knockdown or overexpression was confirmed by immunoblot analysis.

**Measurement of ROS.** ROS levels were measured with OxiSelect In vitro ROS/RNS Assay Kit (Cell Biolabs, San Diego, CA, USA). Starved BMMs (2 × 10$^5$ cells)

were incubated with dichlorofluorescein diacetate (20 μmol L$^{-1}$) for 10 min at 37 °C and washed before stimulation with RANKL (100 ng ml$^{-1}$) for 10 min in incomplete α-MEM. Dichlorofluorescein (DCF) fluorescence (excitation, 485 nm; emission, 535 nm) of 100 μl of lysed cells (0.5% Triton X-100) was analyzed with GeminiEM fluorescence microplate reader (Molecular Devices, Sunnyvale, CA, USA). The cells were fixed in 4% formaldehyde for 10 min for measurement of intracellular ROS levels in DCF-loaded cells. Confocal images were then obtained in an Olympus FV-1000 confocal laser scanning microscope with an Apochromat ×60objective lens (Olympus, Center Valley, PA, USA).

**Flow cytometric analysis.** The mouse BMMs from WT or *DJ-1*$^{-/-}$ mice were stained with anti-RANK (rabbit IgG, SC9072, 1:20, Santa Cruz Biotechnology, Santa Cruz, CA, USA) and CD11b-APC (rat IgG, 17-0117, clon M1/70, 1:100, eBioscience, San Diego, CA, USA) for 1 h. The cells were then stained with FITC-conjugated goat anti-rabbit IgG (F-2765, 1:500, Life technologies, Eugene, OR) secondary antibodies for 30 min. The cells were analyzed with the FACSCalibur flow cytometer (BD Biosciences) and the data were analyzed with FlowJo software, version 10 (Tree star Inc., Ashland, OR, USA). The FACS gating strategy was shown in Supplementary Fig. 1a.

**Determination of bone resorption.** The BMMs ($2 \times 10^4$ cells per well) were incubated on OsteoAssay surface plates (Corning, Tewksbury, MA, USA) for 5 days in α-MEM containing 30 ng ml$^{-1}$ M-CSF and 100 ng ml$^{-1}$ RANKL. In some experiments, BMMs were pre-cultured in α-MEM containing 30 ng ml$^{-1}$ M-CSF and 100 ng ml$^{-1}$ of RANKL for 3 days, the cells were used as mature OCs (mOCs). To determine bone resorption activity of mOCs from WT or DJ-1 KO BMMs, equal number of mOCs ($2 \times 10^4$ cells per well) were further incubated on the osteologic disc in α-MEM containing 30 ng ml$^{-1}$ M-CSF and 100 ng ml$^{-1}$ of RANKL for 2 days. The cells were completely removed from the osteologic disc using 6% sodium hypochlorite solution and then the disc was thoroughly washed with distilled water. The area of resorption pits on air-dried disc was measured with Multi Gauge V3.1 software (Fujifilm, Tokyo, Japan).

**Immunoprecipitation and immunoblotting assay.** BMMs were incubated for 5 h in serum-free α-MEM prior to treatment with RANKL (100 ng ml$^{-1}$) or $H_2O_2$ (0.1–0.5 mM) for the indicated time. The cells were then washed twice with ice-cold phosphate buffered saline (PBS) and lysed in ice-cold lysis buffer (20 mM HEPES, pH 7.5, 150 mM NaCl, 1% Nonidet P-40, 10% glycerol, 60 mM octyl β-glucoside, 10 mM NaF, 1 mM $Na_3VO_4$, 1 mM PMSF, 2.5 mM nitrophenylphosphate, 0.7 μg ml$^{-1}$ pepstatin, and one protease inhibitor cocktail tablet). The cell lysates were kept on ice for 30 min and then centrifuged at 13,000×$g$ for 10 min at 4 °C. The supernatant fraction was "precleared" with the addition of 50 μl protein A/G–agarose followed by gentle rocking for 3 h. Samples of the precleared supernatant fraction of equal protein content were used for immunoprecipitation experiments. The proteins were immunoprecipitated by overnight incubation (at 4 °C with gentle rocking) with specific antibodies and, in turn, protein A/G–agarose. The agarose was washed five times with a washing buffer (20 mM HEPES, pH 7.5, 150 mM NaCl, 0.1% Nonidet P-40, 10% glycerol, 10 mM NaF, and 1 mM $Na_3VO_4$) and dissolved in ×2 Laemmli buffer. The equal amount of samples was subjected to immunoblot analysis by using specific antibodies against DJ-1 (goat IgG, SC-27004, N-20, 1:200), NFATc1 (mouse IgG, SC-7294, 7A6, 1:200), RANK (mouse IgG, SC-374360, H-7, 1:200 or rabbit IgG, SC9072, H300, 1:200), TRAF6 (mouse IgG, SC-8409, D-10, 1:200 or rabbit IgG, SC-7221, H-274, 1:200), Cathepsin K (CatK, rabbit IgG, SC-30056, H-50, 1:200 or goat IgG, SC-6506, C-16, 1:200), c-Fos (rabbit IgG, SC-52, 4, 1:200), TRAP (goat IgG, SC-30833, K-17, 1:200), Syk (rabbit IgG, SC-1077, N-19, 1:200), and SHP-1 (mouse IgG, SC-7289, D-11, 1:200 or rabbit IgG, SC-287, C-19, 1:200), all of which were from Santa Cruz Biotechnology, as well as antibodies against phosphotyrosine (pY) (mouse IgG, 05-321, clon 4G10, 1:1000), c-Src (mouse IgG, 05-184, clon GD11, 1:1000), FcRγ (rabbit IgG, 06-727, 1:1000), and actin (mouse IgG, MAB1501, clon C4, 1:5000) obtained from Millipore (London, UK). Antibodies against the phospho-Syk (Tyr525/526) (rabbit IgG, #2711S), phospho-Gab2 (Tyr452) (rabbit IgG, #3882S), total Gab2 (rabbit IgG, #3239), phospho-PLCγ2 (Tyr1217) (rabbit IgG, #3871S), total PLCγ2 (rabbit IgG, #3872), phospho-Akt (Thr308) (rabbit IgG, #9275S), total Akt (rabbit IgG, #9272S), phospho-ERK1/2 (Thr202/Tyr204) (mouse IgG, #9106S), total ERK1/2 (rabbit IgG, #9102S), phospho-p38 (Thr180/Tyr182) (rabbit IgG, #9215), total p38 (rabbit IgG, #9212), phospho-JNK (Thr183/Tyr185) (rabbit IgG, #9251L), total JNK (rabbit IgG, #9252S). phospho-IκB-α (Ser32/36) (mouse IgG, #9246S), and total IκB-α (rabbit IgG, #9242S) were from Cell Signaling Technology (Beverly, MA, USA). All antibodies obtained from Cell Signaling Technology were diluted at a ratio of 1:1000. Antibody against calcitonin receptor (rabbit IgG, 250618, 1:100) was purchased from Abbiotec (San Diego, CA, USA). Antibody against OSCAR (Goat IgG, AF1633, 1:100) was from R&D Systems Inc (Minneapolis, MN, USA). Immunoreactive proteins were detected with horseradish peroxidase (HRP)-coupled secondary antibodies (for mouse origin, horse IgG #7076S, 1:2000, Cell Signaling Technology; for rabbit origin, goat IgG, #7074S, 1:2000, Cell Signaling Technology; or for goat origin, donkey IgG, SC-2020, 1:5000, Santa Cruz Biotechnology) and enhanced chemiluminescence according to the

manufacturer's protocol (Amersham Biosciences, Piscataway, NJ, USA). Band intensities were measured by using Multi Gauge V3.1 software (Fujifilm).

**In vitro protein kinase and phosphatase assay.** Immunoprecipitates of Syk (rabbit IgG, SC-1077, N-19, Santa Cruz Biotechnology) from whole-cell lysates (1 mg of protein) were assayed for tyrosine kinase activity with ELISA-based Universal Tyrosine Kinase Assay Kit (Genway). Similarly, SHP-1 immunoprecipitates (mouse IgG, SC-7289, D-11, Cell Signaling Technology) were analyzed for phosphatase activity by using SHP-1 activity kit (Duoset IC ELISA, R&D Systems Inc.) according to the manufacturer's instructions.

**Statistical analysis.** The data were presented as the means ± S.E.M from three or more independent experiments, at least in triplicate, for in vitro cell experiments and from five or more mice for all animal experiments. Statistical analysis was performed by Student's $t$-test or Mann–Whitney test. For comparison between multiple values, one-way analysis of variance (ANOVA) with Tukey's post hoc test to assess differences between specific groups. All statistical calculations (*$p < 0.05$ and **$p < 0.01$) were performed with SigmaStat software version 12 (Systat Software, Inc, Point Richmond, CA).

**Data availability.** The data supporting the findings of this study are available within the article and its Supplementary Information files and from the corresponding author on reasonable request.

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

## Acknowledgements

We are particularly grateful to Dr Michael Beaven, one of the co-authors who passed away during the revision of this paper, to dedicate this paper to him. This research was supported by the National Research Foundation of Korea (NRF) grant funded by the Korea government (MSIP, NRF-2016R1A2B3015840 and NRF-2013R1A4A1069575) and in part by NRF-2016R1A5A2012284. M.A.B. was supported by the NHLBI Intramural Program, National Institutes of Health.

## Author contributions

W.S.C. designed the experiments and wrote the paper. Hyuk.S.K. analyzed the data, and wrote the paper. S.H.M. and S.-K.L. performed bone histology. S.T.N., H.W.K. and Y.H. P. performed in vivo experiments and collected the data. B.K. and K.-J.W. contributed reagents and knockout animals. H.-R.K. analyzed the human sample data. Y.M.P. and

Hyung, S.K. performed all experiments for cell signaling analysis. Y.M.K. and M.A.B. provided intellectual input and wrote the paper.

**Additional information**

**Competing interests:** The authors declare no competing financial interests.

