## [Peer Review file · Nature Communications]

Reviewers' comments:

Reviewer #1 (Remarks to the Author):

Kim et al. investigated a role of DJ-1 in osteoclast differentiation in vitro and in vivo using DJ-1-null arthritis model mice and found that DJ-1 negatively regulates the cascade of osteoclastogenesis through the decreased ROS level, leading to activating SHP-1. The results suggest that DJ-1 participates in bone homeostasis through regulation of ROS levels

Although experiments were well organized and interesting, physiological significance of DJ-1 in bone homeostasis and bone diseases is lacking.

Specific comments

1. Physiological significance of DJ-1 in bone homeostasis and bone diseases is lacking. To do that, different phenotype and characteristics of DJ-1 in patients with bone diseases should be shown.

2. Many controls are lacking as described below.

1) Fig. 2k, 2l and 4e

Expression levels of endogenous and ectopically expressed DJ-1 in wild-type and DJ-1 KO BMMs should be shown by Western blotting.

2) Fig. 3a, 3b and 3e

Expression levels of loading control such as actin in wild-type and DJ-1 KO cells should be shown by Western blotting.

3) Fig. 3d, 4a, 4h and 4j

Expression levels of total Akt, Erk1/2, Jun, IκBα, Sky, PLCγ2 and Gab2 in wild-type and DJ-1 KO cells should be shown by Western blotting.

4) Fig. 4g

Expression levels of DJ-1 and a loading control in wild-type and si-DJ-1-treated cells should be shown by Western blotting.

3. Discussion

Authors just described that a role of DJ-1 in osteoclast differentiation is regulation of ROS levels at the position upstream of RANK (Fig. 8). However, there are many reports showing that DJ-1 directly regulates signaling molecules via direct interaction. For instant, DJ-1 directly interacts with SHP-1 (Kim et al. Neurobiol Dis. 2013), which is an important player in this manuscript. I think that author should also discuss from these points, including SHP-1 and other DJ-1-interacting signaling molecules.

Reviewer #2 (Remarks to the Author):

In their manuscript, Kim et al describe an in vivo osteoporotic phenotype for male DJ-1 ko mice with ROS/SHP-1 playing a key mechanistic role in osteoclasts. Overall the studies are well designed and controlled, especially for the in vitro assays. Examination

of key aspects of the in vitro mechanism in human OC cultures as well as rescue of the KO phenotype with DJ-1 re-expression are strengths as well. Rescue of DJ-1 ko phenotype in vitro with overexpression of SHP-1 would further strengthen the proposed mechanism.

The main weakness of the paper is the interpretation of the CIA experiment with respect to the effect on osteoclasts. In this model, there is significantly more paw swelling, likely indicating more inflammation. Since osteoclastogenesis in this model is secondary to the inflammation, it is not possible to conclude that DJ-1's effects within osteoclasts are responsible for the higher OC number and increased resorption. Since this is a global KO model, it is likely that most inflammatory osteolysis models would suffer from the same problem, and it seems that DJ-1 is an important regulator of ROS in inflammatory cells. Proof of a direct effect on OCs in vivo could perhaps be elicited with RANKL injection (IP or over calvaria). As a more minor point, there are a few inconsistencies with the data shown in Fig 7. The arthritis index shown in panel c has very little variability, but the incidence between 35-50 weeks is low/moderate. Does panel c only include mice considered positive for arthritis in d? Likewise the variability for histological score in panel f is low, and also the overall score is low, indicating a mild arthritis. I presume day 60 was the endpoint for this experiment, but I did not see that stated anywhere. Lastly, NFATc1 and c-fos expression from whole joint lysates cannot be considered to be specific for osteoclasts, as inflammatory cells also should have significant expression and may be more abundant than OCs.

Additional minor points

1. In Fig 1, panel c should show a higher magnification to better show OCs. The results section referring to panel d lists osteoclast number and frequency, but OcS/BS is not really a frequency and should be referred to as osteoclast surface.
2. Since it is not likely that anyone will pursue the OB phenotype further, it would be nice to show here whether the increase in Obs/BS is cell autonomous (is there a difference in vitro) or likely secondary to the increase in OCs (as is seen in high turnover osteoporosis).
3. In my opinion, the female bone phenotype should be added to Fig 1 and not hidden in the supplementary data. There is increased interest in sex differences overall, and therefore the data should be considered important enough to include in the main paper, even if the results are different from the males. Also, the manuscript does not state if all of the subsequent in vitro work is in males, or if in vitro the sex does not matter.
4. Fig 2f does not show actin rings. These are structures only seen on a bone or hydroxyapatite substrate. It seems that what is shown here is really just the staining in mononuclear cells, although the magnification is really too low to tell much. As the decrease in resorption in fig 2h/i parallels almost exactly the increase in OC number in fig 2f/g, it is likely that there is no additional effect of OC activity – only an effect on the generation of the mature cells. Either the text should be modified to remove any conclusion of an effect on activity, or resorption should be normalized to the number of mature cells (or cells can be lifted and replated after differentiation (usually this can be done at day 3) to normalize the number of OCs in the resorption assay. If there is a difference in resorption per cell, then it would be useful to examine the actin rings more carefully, but if not then there will be little yield in that exercise.

We thank the editor and reviewers for their helpful and invaluable comments. In response, we have conducted many additional experiments and made appropriate revisions to address all concerns raised by the reviewers. The changes in the text are noted in red. Our responses (in red text) to the reviewers' individual comments (in italics) are given below.

Reviewer 1

Comment #1: *Physiological significance of DJ-1 in bone homeostasis and bone diseases is lacking. To do that, different phenotype and characteristics of DJ-1 in patients with bone diseases should be shown.*

Response: The reviewer's comment is very reasonable and we intend to address it as they would help improve the quality of the manuscript. However, we realize, that addressing this comment would strengthen the relevance of our data to human disease but we are unable to perform relevant studies in patients because of institutional ethical concerns in obtaining internal bone tissue containing osteoclast. In addition autopsy specimens are not readily available to us. Instead, we have tried to demonstrate the reviewer's concern with the result of intracellular DJ-1 function in the process of human osteoclast differentiation from human CD14⁺ osteoclast precursor cells by using siRNAs against DJ-1 (Fig.2a-2d). It is well accepted that the balance between osteoblast (OB) and osteoclast (OC) function is critical for bone homeostasis in both normal and disease conditions (please see ref #1, 2 in this paper). Our in vivo (Fig. 1 for normal bone metabolism, Fig. 6 and Supplementary Fig. 3 for RANKL or LPS-induced OC formation in vivo) and in vitro (Fig. 2) results indicate that the function of intracellular DJ-1 is critical for regulating bone metabolism under normal and pathological conditions. Additionally, we evaluated the phenotypic changes of secreted DJ-1 in the serum and synovial fluid from patients with rheumatoid arthritis (RA) or osteoarthritis (OA) as the control (Supplementary figure 5). As shown in new Supplementary figure 5, the amount of ROS and secreted DJ-1 were increased in patients with RA compared to OA (Supplementary Fig. 5a and 5b). Particularly, the increase was clear and significant in the synovial fluid (Supplementary 5b). Interestingly, the amount of ROS was higher in female patients compared to in male patients (Supplementary Fig. 5c). However, the concentration of DJ-1 was lower in female patients, compared to in male patients (Supplementary Fig. 5c). These results are consistent with that the prevalence of RA is much higher in women than in men (see ref #3, 51, 52). It is clinically very informative since there is no report of secreted DJ-1 phenotype in bone-related disease. As the reviewer could recognize, the specimens do not include osteoclasts and thus have limitations in directly relating to our conclusions on the role of intracellular DJ-1.

In accordance with the reviewer's concern, we will also consider modifying the title specifically to "Intracellular DJ-1 controls bone homeostasis through the regulation of osteoclast differentiation in normal physiology as well as in bone-associated pathology" with the consent of the reviewers and editor. These results with patients are described in the Methods (page 27, line 5 from the bottom-page 28, line 2) and Discussion section (page 14, line 5 from the bottom-page 15, line 2; page 18, lines 4-13).

Comment #2: *Many controls are lacking as described below.*

1) *Fig. 2k, 2l and 4e: Expression levels of endogenous and ectopically expressed DJ-1 in wild-type and DJ-1 KO BMMs should be shown by Western blotting.*

Response: Based on reviewer's comments, we have added representative images for DJ-1 expression (See Fig. 2k, 2l, and 4e).

2) *Fig. 3a, 3b and 3e: Expression levels of loading control such as actin in wild-type and DJ-1 KO cells should be shown by Western blotting.*

Response: As recommended, we have added representative images for actin as each loading control (See Fig. 3a and 3b) and for target proteins as the input control of immunoprecipitation (See Fig. 3e).

3) *Fig. 3d, 4a, 4h and 4j*

Expression levels of total Akt, Erk1/2, Jun, IkBa, Sky, PLC γ 2 and Gab2 in wild-type and DJ-1 KO cells should be shown by Western blotting.

Response: As suggested, we have added representative Western blot images for each control (See Fig. 3d, 4a, 4h, and 4j).

4) *Fig. 4g*

Expression levels of DJ-1 and a loading control in wild-type and si-DJ-1-treated cells should be shown by Western blotting.

Response: As the reviewer's recommendation, we have added the representative images for expressions of DJ-1 and actin as the loading controls (See Fig. 4g).

Comment #3: *Authors just described that a role of DJ-1 in osteoclast differentiation is regulation of ROS levels at the position upstream of RANK (Fig. 8). However, there are many reports showing that DJ-1 directly regulates signaling molecules via direct interaction. For instant, DJ-1 directly interacts with SHP-1 (Kim et al. Neurobiol Dis. 2013), which is an important player in this manuscript. I think that author should also discuss from these points, including SHP-1 and other DJ-1-interacting signaling molecules.*

Response: We agree with the reviewer's comment. DJ-1 is known to be involved in a variety of intracellular signaling pathways, including those associated with cancer and Parkinson's disease. Within cells, DJ-1 modulates the activity of proteins by regulating ROS concentration or their direct interaction. DJ-1 physically binds to PTEN, MEKK1, p53, Daxx, and Nrf2. In the other hand, DJ-1 can regulate the activity of AMP-activated protein kinase, SHP-1, and SHP-2 by modulating the ROS concentration within cells. Recently, it was reported that DJ-1 functions as a scaffold protein for STAT1 and SHP-1 interactions in the brain microglia and astrocytes. In our experiment, SHP-1, which plays a critical role in OC formation, was inhibited by ROS treatment or in DJ-1 KO cells (Fig. 4b, c). We conducted additional experiments according to the reviewer's

comment, but no association of DJ-1 and SHP-1 by RANKL stimulation was observed (data not shown). Our results suggest that during OC differentiation, DJ-1 regulates the activity of SHP-1 by regulating the ROS concentration in the cells. We have described this in the "Discussion" section according to the comments of the reviewer (page 17, line 10- page18, line 3).

Reviewer 2

Comment #1: *The main weakness of the paper is the interpretation of the CIA experiment with respect to the effect on osteoclasts. In this model, there is significantly more paw swelling, likely indicating more inflammation. Since osteoclastogenesis in this model is secondary to the inflammation, it is not possible to conclude that DJ-1's effects within osteoclasts are responsible for the higher OC number and increased resorption. Since this is a global KO model, it is likely that most inflammatory osteolysis models would suffer from the same problem, and it seems that DJ-1 is an important regulator of ROS in inflammatory cells. Proof of a direct effect on OCs in vivo could perhaps be elicited with RANKL injection (IP or over calvaria).*

Response: We agree that the effect of DJ-1 in these CIA animal models may be likely to be global and secondary. We therefore examined the in vivo effect of direct osteoclast differentiation by injection of RANKL in DJ-1 KO mice as suggested by the reviewer (See the method section for this experiment: page 21, line 4 from the bottom – page 22, line 1). The number of osteoclasts in DJ-1 KO mice was significantly increased when RANKL was injected into calvaria, which does not cause inflammation, compared to that in WT mice (Fig. 6). Additionally, the only cell capable of resorbing bone in bone tissue is osteoclast and the number of osteoclasts was increased in DJ-1 KO arthritic mice compared to WT arthritic mice (Fig. 7g). The amounts of osteoclast-specific bio-marker proteins such as TRAP, cathepsin K, calcitonin receptor, and OSCAR were also increased in the joint tissues of DJ-1 KO arthritic animal models (Fig. 7h,i). As pointed out by the reviewer, it can not be said that the increase in OC in RA was entirely because of the effect of intracellular DJ-1 in OC precursor cells. However, the results of RANKL injection and in vitro experiments suggest that intracellular DJ-1 plays a critical role in OC differentiation in vivo as well as in vitro, and at least partially in RA bone tissue. We described the newly added results (Fig. 6) in Abstract (page 2, lines 3-5 from the bottom), the Results section (page 11, lines 10-16) and rewrote the Discussion section to reflect the comments of the reviewer (page 18, line 3 from the bottom – page 19, line 11).

Comment #2: As a more minor point, there are a few inconsistencies with the data shown in Fig 7. The arthritis index shown in panel c has very little variability, but the incidence between 35-50 weeks is low/moderate. Does panel c only include mice considered positive for arthritis in d? Likewise the variability for histological score in panel f is low, and also the overall score is low, indicating a mild arthritis.

Response: We followed the protocol described in the paper "Nature protocol 2007; Vol. 2, No. 5, pages 1269-75 and Arthritis Res Ther 2007; Vol. 9, No. 5, page R113" to induce arthritis in C57BL/6 background mice. Typically, arthritis symptoms and histology scores in C57BL/6 background mice are relatively mild compared to in DBA/1J background mice, which is the

typical mouse used as an arthritis model. Our results were presented as the "mean \pm standard error (SEM)" and therefore the reviewer may feel that the variation of the results is somewhat less. According to the protocol, the arthritic prevalence is judged to be "positive" when it is above a certain arthritic index. Thus, the arthritis index and disease prevalence may appear to be different. We described the evaluation of the prevalence of arthritis in the "Methods (page 22, line 1 from the bottom-page 23, line 1). To address the concerns of the reviewer, we repeated the experiments. We also found that the severity and incidence of arthritis were generally similar to those of previous experiments. Based on the additional results, the results were slightly modified in the severity of arthritis and the cumulative incidence of disease (see Fig. 7c and 7d).

Comment #3: *I presume day 60 was the endpoint for this experiment, but I did not see that stated anywhere.*

Response: As the reviewer pointed out that the endpoint of the experiment was missing, changes have been incorporated in the Methods (Page 23, lines 2-3). On the 60th day after the start of the experiment, the mice were euthanized and then the following experiments were conducted.

Comment #4: *Lastly, NFATc1 and c-fos expression from whole joint lysates cannot be considered to be specific for osteoclasts, as inflammatory cells also should have significant expression and may be more abundant than OCs.*

Response: We fully agree with the reviewer. To overcome, we evaluated more typical mature osteoclast-specific marker proteins. The results confirmed that the amounts of TRAP, cathepsin K, calcitonin receptor, and OSCAR were greatly increased in DJ-1 KO mouse joint tissues (see Fig. 7h and 7i) compared to WT controls. We have described the new results in the Results section (page 12, lines 1-5 from the bottom).

Comment #5: *In Fig 1, panel c should show a higher magnification to better show OCs. The results section referring to panel d lists osteoclast number and frequency, but OcS/BS is not really a frequency and should be referred to as osteoclast surface.*

Response: We magnified the image by 100-fold (Fig. 1e). We also corrected the error in the description of OcS/BS as described in the Results (page 6, lines 11-13).

Comment #6: *Since it is not likely that anyone will pursue the OB phenotype further, it would be nice to show here whether the increase in ObS/BS is cell autonomous (is there a difference in vitro) or likely secondary to the increase in OCs (as is seen in high turnover osteoporosis).*

Response: We appreciate the advice of the reviewer. We have performed the additional experiments for OB differentiation from osteoblastic stromal cells isolated from DJ-1 KO or WT mice. The results showed that OB differentiation was, unexpectedly, slightly reduced in DJ-1 KO cells, compared to WT cells (See Supplementary Fig. 4). These results suggest that the increase of osteoblasts in vivo did not occur because of the cell autonomous increase but rather because of the secondary feedback mechanism to bone loss in DJ-1 KO mice. We have described this in the

Discussion (page 14, lines 4-9).

Comment #7: *In my opinion, the female bone phenotype should be added to Fig 1 and not hidden in the supplementary data. There is increased interest in sex differences overall, and therefore the data should be considered important enough to include in the main paper, even if the results are different from the males. Also, the manuscript does not state if all of the subsequent in vitro work is in males, or if in vitro the sex does not matter.*

Response: We agree with the reviewer's suggestions. We now showed the results from the female bone density in Fig. 1 and described the results in the Results section (page 6, lines 5-6). In addition, we discussed the differences in the incidence of male and female patients and their relevance to DJ-1 in the Discussion section (page 14, line 10- page 15, line 2). We used cells from male mice for all in vitro experiments and clarified it in the Methods section (page 20, Lines 6-7).

Comment #8: *Fig 2f does not show actin rings. These are structures only seen on a bone or hydroxyapatite substrate. It seems that what is shown here is really just the staining in mononuclear cells, although the magnification is really too low to tell much.*

Response: We have carried out additional experiments to obtain better images based on the reviewer's comment and replaced the image with a new image (Fig. 2e). The results indicate that the actin rings were more prominent in large OCs in DJ-1 KO cells than in WT OCs.

Comment #9: *As the decrease in resorption in fig 2h/i parallels almost exactly the increase in OC number in fig 2f/g, it is likely that there is no additional effect of OC activity – only an effect on the generation of the mature cells. Either the text should be modified to remove any conclusion of an effect on activity, or resorption should be normalized to the number of mature cells (or cells can be lifted and replated after differentiation (usually this can be done at day 3) to normalize the number of OCs in the resorption assay. If there is a difference in resorption per cell, then it would be useful to examine the actin rings more carefully, but if not then there will be little yield in that exercise.*

Response: According to the recommendation, we normalized the number of cells according to the suggested protocol and analyzed a bone resorption. For the two different mature osteoclasts from DJ-1 KO or WT cells, each bone resorption activity was nearly identical (Fig. 2g and 2h). These results suggest that intracellular DJ-1 is involved in OC formation, but not in bone resorption activity of mature OCs. Based on these results, we modified the description of the results to state that DJ-1 controls the differentiation process of OC but does not affect the activity of mature OC (Page 7, lines 2-3 from the bottom) and added the procedure for the experiment in the Methods section (page 28, lines 2-7 from the bottom).

All authors thank very much again both reviewers for pointing out weaknesses of this manuscript and offering valuable suggestions. These were most helpful.

Reviewers' comments:

Reviewer #1 (Remarks to the Author):

In the revised manuscript, almost all of my comments have been cleared and authors' reply is acceptable. Now I think that this manuscript is at the acceptable level.

Reviewer #2 (Remarks to the Author):

The paper has been extensively revised, strengthening and clarifying the conclusions. However, there are a few minor issues remaining.

1. Fig 2e shows actin belts on glass, not the actin rings that define resorptive structures on bone or HA. As I said before, only resorbing OCs have actin rings. It is OK to revise the text to say "actin belt" and clarify the substrate, rather than to repeat the experiment, since the pit assay shows no difference in resorption per OC.
2. The images in Fig 2g appear to show counterstained nuclei rather than pits, although it is not possible to tell for certain without any scale bars. The methods don't indicate any form of staining was done prior to cell removal, but unstained osteologic discs would not appear blue/purple. It is not absolutely necessary to show the images of the pits, but if included, the images must correctly reflect the experiment described.
3. The experiment with the human synovial fluid is very difficult to reconcile with the rest of the paper, especially with the addition to the title for "intracellular DJ-1." I understand the attempt at clinical relevance, but the inclusion of this data only in the discussion is not appropriate. The point being made, especially with respect to the sex difference, is really not clear. I would recommend leaving this data out and constructing a separate study in patients or mice to understand what is going on. It is possible that the DJ-1 in synovial fluid is simply released from damaged cells, and has most of its effect intracellularly, but probably a separate experiment with injection of recombinant DJ-1 in the CIA model would be needed. That is beyond the scope of this paper.
4. I do not understand this sentence from the bottom of page 14: "In humans, these are overwhelmingly higher in women when compared with the prevalence of osteoporosis and RA in men of age similar to that of menopausal women 3,51,52." Please revise this and indicate what "these" refers to.
5. Also on p 14, I think you are trying to say that you do not see effects in female mice at baseline because the WT bone mass is already very low, so perhaps you could not detect a further decrease. You should not say that this is "reduced largely" or "decreased to the level in DJ-1 ko male mice" -bone mass is just lower in females than in males (ie the peak bone mass attained in females is lower than males). Did you ever compare DJ-1 levels in male vs female mice? I gather from the human data that you think perhaps females have less DJ-1 overall, which maybe could contribute to more OC differentiation. You might actually be able to detect effects on DJ-1 loss in the females in a pathological model, even if there is no difference at baseline (I have observed this in our own work at times), or it is possible that females simply do not use DJ-1 as a negative regulator of OCs. I do not think you need to do additional experiments here, but this new paragraph on the sex difference needs to be clarified and could include some ideas for future work to address the issue.

We thank the editor and reviewers for their helpful and invaluable comments. In response, we have made appropriate revisions to address the concerns raised by the 2nd reviewer. The changes in the text are noted in red. Our responses (in red text) to the reviewers' individual comments (in italics) are given below. The line number in the response means those on the pdf created by the Nature submission system.

Reviewer 1

Comment #1: *Fig 2e shows actin belts on glass, not the actin rings that define resorptive structures on bone or HA. As I said before, only resorbing OCs have actin rings. It is OK to revise the text to say “actin belt” and clarify the substrate, rather than to repeat the experiment, since the pit assay shows no difference in resorption per OC.*

Response: In accordance with the reasonable comment from the reviewer, we modified the "actin ring" to "actin belt" in Fig. 2e. The text has also been modified accordingly in the sections for the “Results” (line 119), “Methods” (line 430), and “Legends” (line 783). We cultured BMMs in a 96-well clustered plastic culture plate for 4 days to identify the actin belt formation of OC (line 427).

Comment #2: *The images in Fig 2g appear to show counterstained nuclei rather than pits, although it is not possible to tell for certain without any scale bars. The methods don't indicate any form of staining was done prior to cell removal, but unstained osteologic discs would not appear blue/purple. It is not absolutely necessary to show the images of the pits, but if included, the images must correctly reflect the experiment described.*

Response: To address the problem posed by this reviewer, we repeated the experiment in slightly different ways without staining. OCs were cultured on an osteologic disc and completely removed by 6% sodium hypochlorite solution, and finally the bone resorption area of the air-dried disc was measured without staining by toluidine blue. We replaced the previous images with newly obtained images (Fig. 2g) and the new quantitative data (Fig. 2h), which were very similar to the previous ones. We also clarified the practical procedure in the section for “Methods”(lines 497-499) and “Legends” (lines 785-786). The scale bar was also inserted in Fig. 2g as recommended by the reviewer.

Comment #3: *The experiment with the human synovial fluid is very difficult to reconcile with the rest of the paper, especially with the addition to the title for “intracellular DJ-1.” I understand the attempt at clinical relevance, but the inclusion of this data only in the discussion is not appropriate. The point being made, especially with respect to the sex difference, is really not clear. I would recommend leaving this data out and constructing a separate study in patients or mice to understand what is going on. It is possible that the DJ-1 in synovial fluid is simply released from damaged cells, and has most of its effect intracellularly, but probably a separate experiment with injection of recombinant DJ-1 in the CIA model would be needed. That is beyond the scope of this paper.*

Response: We agree with the comments of this reviewer. As per the reviewer's comments, the DJ-1 in synovial fluid and serum is, most possibly, released from damaged cells, and has most of its effects intracellularly, thus having limitations in directly relating to our conclusions. In response to the reviewer's comment, we would like to remove the results (Supplementary Fig. 5) obtained from the patient samples. In addition, we would also like to remove "intracellular" from the title of the article, as recommended by the reviewer, so that we can convey a clearer message to more readers.

Comment #4: *I do not understand this sentence from the bottom of page 14: "In humans, these are overwhelmingly higher in women when compared with the prevalence of osteoporosis and RA in men of age similar to that of menopausal women 3,51,52." Please revise this and indicate what "these" refers to.*

Response for comments 4: The sentence was properly rewritten to reflect the comment of this reviewer (lines 258-259).

Comment #5: *Also on p 14, I think you are trying to say that you do not see effects in female mice at baseline because the WT bone mass is already very low, so perhaps you could not detect a further decrease. You should not say that this is "reduced largely" or "decreased to the level in DJ-1 ko male mice" –bone mass is just lower in females than in males (ie the peak bone mass attained in females is lower than males). Did you ever compare DJ-1 levels in male vs female mice? I gather from the human data that you think perhaps females have less DJ-1 overall, which maybe could contribute to more OC differentiation. You might actually be able to detect effects on DJ-1 loss in the females in a pathological model, even if there is no difference at baseline (I have observed this in our own work at times), or it is possible that females simply do not use DJ-1 as a negative regulator of OCs. I do not think you need to do additional experiments here, but this new paragraph on the sex difference needs to be clarified and could include some ideas for future work to address the issue.*

Response for comments 4 and 5: Based on the reasonable comments of this reviewer, we have added some discussion about the sex difference of bone phenotype by DJ-1, its clinical implications, and some future direction (lines 245-265).

REVIEWERS' COMMENTS:

Reviewer #2 (Remarks to the Author):

Thank you for modifying the manuscript carefully. All of my concerns have been addressed.